# Religious Amnesias, Mythologies, and Apolitical Affects in Racist Landscapes

**Sunder John Boopalan**

Department of Biblical and Theological Studies, Canadian Mennonite University, Winnipeg, MB R3P 2N2, Canada; JBoopalan@cmu.ca

**Abstract:** Given their explicit attention to contextual realities, liberation theologies have different expressions in various global contexts. One element they all have in common, however, is a sustained interest in the effects of historical processes. Dalit theology, a liberation theology arising from the struggles and hopes of Dalit communities in India, is in attunement with such critical analyses of the factors that shape power and domination. By drawing comparisons between the geography of a typical Indian village/town—in which bodies are segregated by caste belonging—and the increasing gentrification in towns and cities in the U.S.—in which bodies are segregated by the aftereffects of racialized geographies—the essay argues that domination today is better understood through affective encounters or the lack thereof.

**Keywords:** gentrification; race; caste; affect; church; segregation; happiness; positive psychology; amnesia; mythology; racism

---

## 1. Introduction

Seated at the back row of a large classroom as a teaching assistant for a class of over 50 students in the U.S., I watched student reactions to the question posed by guest lecturer and author of the book, *The Christian Imagination*, Willie Jennings. Jennings asked, "How many of you know what a 'sundown town' is?" Some hands went up but, overall, less than 10% of the class knew what a "sundown town" was. Although such towns sometimes included exclusion of Native Americans and other people of color such as Mexican Americans, Chinese Americans, and Jews, the name "sundown town", as James E. Loewen explains, derives from a typical sign that stood outside many of these towns that read, "N*****, Don't Let The Sun Go Down On You In ___".[1] There are no legally sanctioned sundown towns today, but similar exclusionary logics of whiteness[2] permeate the American landscape and increasing racialized segregation today.

In his book, *Color of Law*, Richard Rothstein, after noting that "until the last quarter of the twentieth century, racially explicit policies of federal, state, and local governments defined where whites and African Americans should live", goes on to rightly argue that "today's residential segregation in the North, South, Midwest, and West is not the unintended consequence of individual choices and of

---

1    (Loewen 2009, p. 23) Loewen notes on the same page that "a town or county with very few African American households decade after decade, or with a sharp drop in African American populations between two censuses" may be termed a "sundown town" if it can be shown that such "absence is intentional". Bowdlerization is intentional.

2    Ki Joo Choi, in his book, Disciplined by Race, builds on Melanie Bush's definition to note the following two characteristics of whiteness: "First, the 'compilation of institutional privileges and ideological characteristics bestowed upon members of the dominant group in societies organized by the idea and practice of pan-European supremacy.' Second, the corresponding 'white racial frame' or the 'use of stereotypes, metaphors, images, emotions, and narratives' that 'both emanate from and support [the] systemic racism [of whiteness].'" See (Choi 2019); see also (Bush 2011).

---

otherwise well-meaning law or regulation but of unhidden public policy that explicitly segregated every metropolitan area in the United States".[3]　And yet, despite this well documented reality, Rothstein soberingly observes that "with very rare exceptions, textbook after textbook adopts the same mythology" about segregation today as "just the way things are" or because of some other benign thing.[4] This public portrayal of an otherwise malevolent process has the consequence of sustaining an apolitical view of history and furthering apathy.

I apply Rothstein's analysis of the portrayal of segregation in U.S. textbooks to argue that other public platforms (churches, in this case) intentionally or otherwise further similar mythologies that further political and emotional—as the essay will highlight in subsequent sections—apathy. By analyzing racism through geography/neighborhood and affect, I highlight how domination today is better understood through affective encounters and/or the lack thereof, especially as a result of increasing racialized geographical segregation. An underlying claim is that religions become complicit in racism when they aid problematic affective encounters and/or accentuate political apathy.

The essay also builds on my experience of serving for over three years in pastoral roles in the United States. In employing this method, the essay uses anecdotes, but with names of people and places changed or unnamed. The anecdotes will serve as windows into themes that facilitate greater understanding of racism and religion.

## 2. Religious and Other Affects

My social location as a Dalit theologian means that I bring, among other things, the lens of caste to interrogate domination. Since my coming to the U.S. and having begun a more intentional comparison of caste and race, I have had to expand and deepen my toolbox to understand domination today. Casteism and racism are systems of domination that depend on a complex variety of interdependent factors. I describe the connections between caste, religion, and affect in the Indian context and use my learnings therein to comment on similar entanglements in the U.S., thus moving from caste to race. These comparisons enable a more comprehensive understanding of the problem at hand, across borders.

Consider this brief description of the geography of a typical Indian village/town in which bodies are segregated by caste belonging. In India, when one uses the word "village" in the local vernacular, its meaning is not straightforward. "Village" in fact has *two* meanings. A first meaning, somewhat benign, conveys that the village is a small municipality of a larger district or state. A second level of meaning—one that often escapes the uncritical hearer—refers to *that part of the village* where dominant caste persons live. In the dominant caste understanding, the "village proper" is only that part in which the dominant castes live.

That part of the village where Dalits live, by some act of dominant caste imagination, does not count. The Dalit side of every village is often referenced through various othering terms. The idea of "our side" and "their side", by virtue of being part of the geography of the village, is also deeply inscribed into patterns of thinking (epistemological) and acting (tactile) that constantly assess self and other. Gandhi, who comes from a dominant caste, lifted up the Indian village as an example of great local economy. Ambedkar, Gandhi's ideological opponent, recognizing Gandhi's caste-based blind spots, criticized the Indian village as a *den* of "narrowmindedness".[5]

Because this "us–them" difference is so entrenched, crossing borders of caste is often met with negative, including lethal, affect. In 2015, Sagar Seghwal, a Dalit, was in a public place as his cellphone began to ring. The ring tone was a song that praised B. R. Ambedkar. When dominant caste patrons heard the song, it *affected* them. The affect was so strong that they pulled the boy out and beat him to

3　(Rothstein 2018, pp. 10–11).
4　(Rothstein 2018, p. 359).
5　B. R. Ambedkar, cited in (Boopalan 2017, p. 33).

death. As violent and tragic as Sagar's death may be, it is vital to examine the initial strong *affect* that the dominant caste subjects *felt*. The afteraffects of caste-informed geographies have lethal ramifications.

Casteism in India operates both overtly and covertly. Overtly, as we considered above, casteism in India includes lynchings and murders (by members of dominant castes) of those deemed to violate dominant caste norms. Covertly, caste-based oppression operates through many means including endogamous marriages, caste-based political loyalties, and seemingly benign actions whitewashed under the label "culture".[6]

As Dalit theologian and pastor Deenabandhu Manchala puts it, "my engagement with anti-caste movements has taught me that caste is like a hydra-headed dragon with incredible capacities to survive multiple attacks, and to mutate, assuming more fierce and lethal manifestations". This quote comes from a recent essay in which, referring to how casteism and racism affect Dalits in India and Black people in the U.S., Manchala insightfully observes that "*both their struggles against discrimination are seen as their 'problems,' and not as a moral challenge to the wider society*".[7]

Churches and other religious institutions are often complicit in caste-based oppression (and racism, as subsequent sections show) as well. The complicity occurs when, as Manchala notes, casteism and racism are seen as "their problems" rather than as moral problems that affect us all and that Christians should treat as a core matter pertaining to faith. Instead, churches often participate in causing religiously induced amnesias, mythologies, and apolitical affects.

Let me offer an example of a mythology—more examples further along in the essay—I encountered in a church setting. This is an encounter I keep returning to. I visited an Indian American church congregation in the Boston area. I sat at a table with five others during fellowship hour and someone asked me what my research interests were. "Remembering wrongs", I said. A medical doctor who was at the table interjected and said, "Oh, everyone has a complaint these days. Everyone has a grouse. Everyone is a victim". The doctor was visibly upset. After squirming in his seat for a while, he could not hold it any longer. He butted in, asking a rhetorical question. "Who is the voice of God in America?" He asked for second time, "Who is the voice of God in America?"

In my mind I was thinking, if God is portrayed in Hebrew Scriptures as speaking through a donkey to Balaam, God can speak through anyone, right? So, I gave him a considered theological response. "It could be any of us", I said. The doctor was dissatisfied with this. After rhetorically asking, "Who is the voice of God in America?", he went on to state, "It is a *Black guy* (with emphasis), Morgan Freeman. The voice of God in Hollywood is a *Black guy* and you still want to complain? It's all about one's attitude".

All I said was "remembering wrongs" but it set off something in the doctor. It elicited a strong affect that was then verbalized. The medical doctor's affect prevents him from seeing how his presumably innocent (from his perspective) statement can, in fact, be humiliating to and disrespectful of the experiences of a great number of people who continue to suffer racialized discrimination. Furthermore, isn't the act of remembering wrongs an initial condition of faith in Jesus? Isn't the memory of Jesus necessarily a "memory of the one who was murdered for defying oppressive systems and cultures"?[8] What is it about church spaces that creates amnesia of not only this brute fact of Christian faith but also amnesia of the historical conditions that have given rise to the problems that many think are "just the way they are" or "they've always been this way"? Churches are complicit in furthering not simply the mythology that Jesus died an apolitical death but also mythologies about post-racial America.

---

6    For a more elaborate discussion of these matters, see chapter two, (Boopalan 2017).
7    (Manchala 2020). Emphasis mine.
8    Ibid.

### 3. Segregated America's Forgotten Secrets: "We Come to Church for Diversity"

In the next section, I will return more directly to how churches participate in causing religiously induced amnesias, mythologies, and apolitical affects. Before that, however, let us return to what the essay started with—racialized segregation. Racialized segregation is racist America's forgotten secret.

In a church I was part of, I asked several people during one fundraising season, "Why do you come to this church?" I got several responses but one particular one stood out to me because of its connection to racism and religion. A white member said to me, "My children go to a school that is white. We live in a neighborhood that is not dissimilar. We come to church so our children can experience diversity". Coming to church "for diversity" in America is hard to wrap one's mind around for several reasons. First and fundamentally, as considered in the section above, it buys into the false *de facto* nature of segregation in the country and lifts up white innocence.

In many cities in the U.S. the move from inner cities to suburbs is often made by white and other racially dominant subjects. When such folks move from cities to suburbs, they almost never attribute their intentions to racist logic. From the perspective of the suburbanites, the us–them racial difference, if acknowledged, is often stated as merely geographical. Such crossovers, however, are built upon racialized geographies inherited from the past and further sustain racialized borders in the present. This observation by Rothstein offers helpful context:

> When from 2014 to 2016, riots in places like Ferguson, Baltimore, Milwaukee, or Charlotte captured our attention, most of us thought we knew how these segregated neighborhoods, with their crime, violence, anger, and poverty came to be. We said they are "de facto segregated", that they result from private practices, not from law or government policy. De facto segregation, we tell ourselves, has various causes. When African Americans moved into a neighborhood like Ferguson, a few racially prejudiced white families decided to leave, and then as the number of black families grew, the neighborhood deteriorated, and "white flight" followed. Real estate agents steered whites away from black neighborhoods, and blacks away from white ones. Banks discriminated with "redlining", refusing to give mortgages to African Americans or extracting unusually severe terms from them with subprime loans . . . All this has some truth, but it remains a small part of the truth, submerged by a far more important one: until the last quarter of the twentieth century, racially explicit policies of federal, state, and local governments defined where whites and African Americans should live. Today's residential segregation in the North, South, Midwest, and West is not the unintended consequence of individual choices and of otherwise well-meaning law or regulation but of unhidden public policy that explicitly segregated every metropolitan area in the United States. The policy was so systematic and forceful that its effects endure to the present time. Without our government's purposeful imposition of racial segregation, the other causes—private prejudice, white flight, real estate steering, bank redlining, income differences, and self-segregation—still would have existed but with far less opportunity for expression. Segregation by intentional government action is not de facto. Rather, it is what courts call de jure: segregation by law and public policy.[9]

Despite this well documented reality, Rothstein notes that "With very rare exceptions, textbook after textbook adopts the same mythology"[10] about segregation today as just the way things are and not as planned and executed by public governmental policy. These textbooks are the same textbooks that students in middle schools and high schools are reading in the U.S. today. How do such "false histories"[11] condition public affect? When false histories accentuate forgotten secrets, innocence is

---

9　(Rothstein 2018, pp. 9–11).
10　(Rothstein 2018, p. 359.)
11　Ibid.

simply not available to be claimed. And, yet, this impossibility is performed by white people and their allies in the pursuit of a mistaken and violent innocence.

Cities such as Ferguson, Baltimore, Milwaukee, or Charlotte from 2014 to 2016. And, now in 2020, cities such as Minneapolis and countless others across the United States. The title of novelist Celeste Ng's book, *Little Fires Everywhere* (recently produced as a TV series by the same name), has become both a metaphor for and reality of our time. In the last days and weeks at the time of this writing, there are little fires burning *everywhere*. People who have long suffered racist violence are understandably angry. Like the prophet Jeremiah in Hebrew scripture, they condemn the familiar and everyday experience of "violence and destruction" (Jeremiah 20:8). If one pays attention to their voice, one begins to hear the familiar and haunting words of Jeremiah, "there is something like a burning fire shut up in my bones; *I am weary with holding it in, and I cannot*" (Jeremiah 20:9). One might expect that church goers who are readers of scripture would understand such legitimate anger of those who bear the weight of racist violence. However, many Christians fail to understand such anger and end up coercing people into a false happiness.

Churches participate in racist violence when they simply accept the affects that accompany white innocence. The same church mentioned in this section is a church that describes itself as "diverse" in its self-presentation to its predominantly white members. However, if diversity hinders political awakening and evokes, in its place, apolitical affect, churches end up becoming not much different from a cruise ship experience to foreign and diverse lands that caters to white passengers who want to "experience diversity" not available in their daily so-called ordinary lives. Sunday, in this way, becomes an extraordinary cruise ship experience where they can introduce their children to different others once a week, much like visitors entering the Aussie Aviary at Franklin Park Zoo in Boston where they can "experience" close contact with brightly-colored birds that they don't see every day.

Segregated America today is a consequence of racist public policy and practice. Governments, banks, and police maintain these geographical divisions in overt and covert ways. Referring to such practices in the context of the 2020 protests in Minneapolis over the death of George Floyd who cried "mama" as his neck was throttled, Myron Orfield and Will Stancil note how.

"Lake Street, a major commercial area near the site of Mr. Floyd's death" is a geographical site that happens to separate majority-white residential neighborhoods from racialized others.[12] Such a state of affairs is commonplace in many American towns and cities. By leaving such racialized inequalities unnamed and unaddressed, churches participate in maintaining the secret of segregated America. Furthermore, in yielding themselves as places where white families come "for diversity", churches inadvertently or otherwise participate in sustaining a violence masked as white innocence.

"White innocence" enables us to understand how racism endures in modern America, especially in relation to "a range of geographically grounded practices"[13] rooted in segregationist logic. White identity's connection to hierarchized geography, as Joshua Inwood notes, is critical "for the reproduction of white innocence".[14] A major way in which white innocence operates is by segregating geography from "the legacies of racism and racial exploitation" in such a manner that racism today appears "devoid of a larger socio-spatial context".[15] When we apply this to the aforementioned churchgoer's statement that he and his family come to church "for diversity" in juxtaposition with his prefixing that his family lives in a white neighborhood and his children attend a white school, we begin to understand how "it is the innocence which constitutes the crime".[16]

---

[12]  (Orfield and Stancil 2020).
[13]  (Inwood 2018, p. 2).
[14]  (Inwood 2018, p. 3).
[15]  (Inwood 2018, p. 5).
[16]  James Baldwin, cited in Inwood.

Building on psychologies of perception, recent ethnographic studies demonstrate how "organs of perception" are differently "calibrated according to social location".[17] In other words, one's social status and power, determined by belonging to particular geographies conditioned by race and caste, affect the way in which the brain responds to others. In other words, geographical borders of various kinds that separate people based on race and caste are impacting the fundamental ways in which humans feel for each other.[18] Empathy, therefore, can no longer be considered a given feature of collective social life. Coming to church for one hour in a week "for diversity" does not help in engendering empathy in white children. More fundamentally and problematically, such desire for momentary and feel-good-diversity problematically propagates white innocence in the same children who would then be introduced to the myths that gloss over the forgotten secrets of racist American segregation. Consider this quote from Jemar Tisby's *Color of Compromise*:

> Though it would be far simpler to relegate racism to a single region such as the South as the historic site of slavery and the Confederacy, this is simply not possible. The South has often been used as the foil for the rest of America. People in other parts of the country could always look below the Mason-Dixon Line and say, "Those are the real racists". Yet the very conspicuousness of white supremacy in the South has made it easier for racism in other parts of the country to exist in open obscurity. Christians of the North have often been characterized as abolitionists, integrationists, and open-minded citizens who want all people to have a chance at equality. Christians of the South, on the other hand, have been portrayed as uniformly racist, segregationist, and antidemocratic. The truth is far more complicated. In reality, most of the black people who left the South encountered similar patterns of race-based discrimination wherever they went. Although they may not have faced the same closed system of white supremacy that permeated the South, they still contended with segregation and put up with daily assaults on their dignity, and the church contributed to this.[19]

The truths of racist American segregation in both the North and the South are well documented.[20] In choosing to live in a social arrangement in which borders segregate them from racialized others, white persons and their allies from dominant social locations inculcate in themselves a present and future inability to feel response-able or positive affect for racialized others. Racialized geographies that further segregation today "result in societies where people can't understand each other or work together" and further "reinforce stereotypes and that it erodes people's ability to interact across racial lines".[21] To return to the image of the Aussie Aviary in Franklin Park Zoo, a weekly visit to feel close to brightly colored birds by purchasing a seed stick is not unlike driving to church and putting a little something in the offering plate so white children can "experience diversity" from up close. Both situations enable the subject to feel good momentarily and perpetuate a naïve yet violent innocence. In so far as the church perpetuates this condition without altering such affects, it participates in amnesias, mythologies, and apolitical apathy.

## 4. Apolitical Pursuits of "Happiness": "I Come to Church for Comfort"

Churches are often located in parts of towns and cities that have ambiguous histories with racism. Churches, therefore, are excellent places for engendering conversation on and conversion from racism. In foregoing an analysis of power in the pursuit of "happiness", however, churches participate in creating apolitical apathy. Political apathy is common in churches. If you've been in churches or around Christians long enough, you are bound to have heard some version of "I don't like to hear

---

[17] (Lee 2015, p. 46).
[18] For more on the subject, see (Lee 2015)
[19] (Tisby 2019, p. 129).
[20] See, for instance, (Blum 2005; Purnell 2019; Goldschmidt and McAlister 2004).
[21] (Orfield and Stancil 2020).

about politics at church; I come to church for comfort and peace". The one wound that has recently revealed itself conspicuously and in all its ancient ugliness is the wound of racism. Racism is an old wound that has never been healed because it is has been covered up for too long. One of these covers is a false piety that produces political apathy.

In a two-week timespan in May–June 2020, two white church members from opposite sides of the American partisan political spectrum, told me that they come to church for comfort and that they find the political tone and content of Sunday worship off-putting. I found it extremely curious that in an era where the partisan political divide is extreme, there was agreement across the divide, as it were, on the need for an apolitical church. "I come to church for comfort", they said. A desire for comfort without talking about politics leads to a private, isolated, and secluded form of social practice that does not dialogue with larger social structures and remains passive in relation to surrounding social evil. In this section, I analyze this desire for apolitical comfort in church using, among others, the works of Sarah Ahmed and Oksana Yakushko.

Church leaders and members who, instead of excavating the life sources of racism, traffic in "happiness", may be said to be perpetrating what Yakushko calls "happy hateful states".[22] They are called thus because they lift up happiness at the cost of being complicit in maintaining otherwise hateful structures hidden under the veneer of white innocence. Present herein is a disturbing connection to the dominant field of positive psychology in the U.S. "Positive psychologists", as Yakushko argues "claim that racism and other forms of oppression will disappear when people pursue their own happiness".[23]

Yakushko builds on the work of Sarah Ahmed who, in her book, *The Promise of Happiness*, "elucidates how white people mobilise happiness in expansive ways to tone down or silence antiracist protest and political controversy". The insight from Ahmed that Yakushko excavates is the idea that "if something 'is good', 'it feels good.'"[24] In other words, if talking politics during Sunday morning does not feel good, then it is deemed not good and in need of avoidance or expulsion. Any description of the "marginalisation of the oppressed is heard/interpreted", in this light, "as something that in itself creates unhappiness and misery".[25] Predominantly white churches and/or churches that center whiteness are unable to be part of healing the racial wound because of an unwillingness to air the wound in their apolitical pursuit of "happiness".

There is another crucial piece that is to be recognized in this connection. On the same Sunday that disturbed the happiness of the aforementioned two members, other members—who otherwise opined that a discussion on racist violence was merited at church and participated actively in it—were nevertheless swayed by the affect of these two members and moved from naming racist brutality to mentioning and lifting up "positive examples". Here, Alfred Frankowski's astute observation of how dominant racial subjects feel compelled to talk of "post-raciality" in a public discussion of racism enables one to recognize the problem in the movement *from* describing racist brutality *to* lifting up such "positive examples". Frankowski notes how "to publicly discuss race one is almost obliged to recognize that our present has made progress from the past". In this way "positive examples" of progress in their pursuit of apolitical happiness silence "political forms of remembrance". In the end, "habitual and common ways of thinking about racism become equal to so many ways of failing to think about racism".[26]

Martin Luther King Jr.'s sober reminder in his "Letter from Birmingham Jail" is insightful here. Racism is "like a boil that can never be cured as long as it is covered up but must be opened with all its pus-flowing ugliness to the natural medicines of air and light, injustice must likewise be exposed, with all of the tension its exposing creates, to the light of human conscience and the air of national

---

22　(Yakushko 2019, p. 129).
23　Ibid.
24　(Yakushko 2019, p. 86).
25　Sarah Ahmed, cited in (Yakushko 2019, p. 86).
26　Alfred Frankowski, cited in (Boopalan 2017, p. 97).

opinion before it can be cured".[27] Yakushko and Ahmed say something similar. "Our focus must remain on unhappiness", says Yakushko, "as well as the realities of those who are" in Ahmed's words, "banished from it [happiness] or who enter this history only as troublemakers, wretches, strangers, dissenters, killers of joy".[28] The question that Yakushko adapts from Ahmed, then, is one that churches and religious persons are to consider. "Who defines what happiness is and for what social purpose"?[29]

Defining happiness as the absence of "being too negative" is traced by Yakushko to the very founder of positive psychology, Martin Seligman. Yakushko highlights how Seligman, in his book *Authentic Happiness*, derides "contemporary American demagogues who play the race card, invoking reminders of slavery ... at every opportunity, [and thus] create the same vengeful mindset in their followers".[30] But the question raised in the previous paragraph still remains. Who defines what happiness is and what social purpose does such a definition serve? When this lens is used to analyze responses to the naming of racialized violence, one begins to understand the social function of happiness. To return to the example of the Indian American doctor at church stated at the beginning of the essay, we begin to see that the doctor's mindset in rooted in the idea that happiness is a state of mind that people can will themselves into—something that is done through the exclusion or trivialization of social violence. Such apolitical pursuits of happiness exclude redress of violent structures and practices.

Churches do participate in creating the conditions for apolitical pursuits. Churches often support dominant "social compliance" in ways that pressurize persons "to concede their own perceptions while replacing them with required 'positive' responses".[31] Racist violence is consequently concealed and covered up. Such concealment in pursuits of apolitical happiness may be considered even as spiritual coercion. "Coercion", as Ahmed notes, is often thought of "as an external force that requires the obedience of subjects through the use of threats, intimidation, or pressure".[32] The matter, however, is far more complicated. Coercion into happiness can happen also in soft ways. This is what Ahmed calls "immanent coercion, a demand for agreement".[33] This is evident when we consider how the Sunday morning discussion mentioned above moved from naming racist brutality to mentioning and lifting up "positive examples". Even those who named instances of racist brutality were swayed by such soft coercion.

## 5. Whither from Here?

Religious people are to be deeply aware of such shifts enacted by apolitical pursuits of happiness. Apolitical pursuits of happiness have very real connections with white supremacist thinking.[34] Apolitical pursuits also betray the gospel of a very political Jesus[35] who did not preach that a trip to the beach would get people out of their funk. The notion that interpretation of religion is to be fundamentally apolitical and the corresponding demand "that a presentation of religious or philosophical material should be free of political assessments"[36] represent a curious state of affairs. Still, denigration of political thinking is commonplace within Christian circles and causes apolitical affects that buttress a demotivation to interrogate societal structures and examine the causes and consequences of social evil.

---

[27]  (Bass 2001, p. 125).
[28]  Ahmed, cited in (Yakushko 2019, p. 185).
[29]  (Yakushko 2019, p. 185).
[30]  (Yakushko 2019, p. 186).
[31]  (Yakushko 2019, p. 187).
[32]  Ahmed, cited in (Yakushko 2019, p. 187).
[33]  (Yakushko 2019, p. 187).
[34]  On this topic see chapter 7, "On Being Pollyanna About Sciences" and chapter 8, "Critics and Critiques of Scientific Pollyannaism" in (Yakushko 2019, pp. 153–79, 181–93).
[35]  (Cone 2011; Althaus-Reid 2000, pp. 3–18; Ahn 2016, pp. 145–62; Turman 2013).
[36]  (Ambedkar 2011, p. xi).

Churches participating in apolitical pursuits are, in a very real sense, anomalies. After all, when we consider the life, teaching, and praxis of Jesus, it is, as Mizo liberation theologian RC Jongte observes, "not possible to leave out the socio-political dimension".[37] If Christian belief and practice fails this "test" and becomes "devoid of commitment to social justice concerns then the problem is not only that such spirituality is inauthentic but is also a misconception of Christian spirituality and discipleship", notes Jongte.[38] Yet still, anomalies persist and contradictions abound.

Despite the emergence of participatory democracies in various parts of the world, structural injustices have metamorphosed and mutated, taking on new forms that are not readily recognizable. Religious spaces can be places where recognition of injustice is fostered through the cultivation of sympathetic and political affect, but not when such spaces often and uncritically participate in abstraction and the creation of apolitical affects. Building on Kevin Lewis O'Neill's SueJeanne Koh, "the manipulations of affect profoundly alters the orientation of bodies and the formation of religiously inflected spaces".[39] More simply stated, *feeling* makes a space theological, not that the space is theological in itself.

Abstraction is a friend of injustice. Abstraction, as Asian American theologian Henry Kuo observes, "can serve as a hiding place for evading difficult, concrete action",[40] thus creating apolitical affects, amnesias, and myths of progress. Over against this, churches are to be repositories of the memories that racism conceals. When this is embraced, Christian spirituality "opens the eyes of our hearts, piercing through the veneer of our idealizations of 'real life' or the 'real world' and allowing us to encounter the reality of suffering and the structures of domination that historical or prevailing narratives can obscure".[41]

In speaking of religious amnesias, mythologies, and apolitical affects in racist landscapes, this essay does not seek to locate structures of domination solely in white spaces or predominantly white churches. While the category of whiteness includes white spaces and persons, whiteness extends its reach into non-white spaces and persons as well. The essay highlighted the encounter with the Indian American doctor in an Asian-American church setting precisely to make the observation that apolitical affects inflected by whiteness occur in myriad settings.

The doctor's strong negative reaction to the mention of structural wrongs is significant in light of this essay's problematization of the felt need to lift up "positive examples" in a discussion about racism. The doctor's negative reaction downplayed the existence of racialized discrimination. One might ask, could we not raise "positive examples" and still name racism? While such a both–and approach could lead to outcomes other than amnesia and possibly improve, sharpen, or expand the conversation across various racialized divides, the far-reaching depth at which apolitical affects operate in racialized landscapes tempers readily available optimisms.

Statements such as "We come to church for diversity" and "I come to church for comfort" are complicit in narratives of progress that conceal racism rather than reveal it. The analysis of those two statements with the background of racialized segregation and positive psychological thinking and the pursuit of an abstracted happiness show how this is indeed the case. In conclusion, I highlight how religious people can address racism by lifting up memories, "dangerous memories"—those memories "in which earlier experiences flare up and unleash new dangerous insights for the present. For brief moments they illuminate, harshly and piercingly, the problematic character of things we made our peace with a long time ago and the banality of what we take to be 'realism'."[42] In other words, churches, if they are to resist complicity in racist landscapes, need to be places where the working of domination

---

[37] Oral interview on 14 June 2020.
[38] Jongte's insight comes in dialogue with liberation theologians who have long prioritized a commitment to the oppressed of the world as a question of faith. In other words, pitting faith against justice and political considerations is a false dichotomy.
[39] (O'Neill 2013).
[40] (Kuo 2019, p. 202).
[41] (Kuo 2019, p. 214).
[42] Johannes Baptist Metz, cited in (Kuo 2019, p. 212).

are revealed rather than concealed, aired rather than covered, and where social tension is sustained for reflection and action rather than allowing an apolitical pursuit of happiness disinvested in justice.

**Funding:** This research received no external funding.

**Conflicts of Interest:** The author declare no conflict of interest.

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
