# Peer review of "Religious Amnesias, Mythologies, and Apolitical Affects in Racist Landscapes"

_religions, doi:10.3390/rel11110615_

Round 1

Reviewer 1 Report

Enjoyed reading the article;

just one question on the "apolitical" and raising "positive example"

Could raising "positive examples" while naming racism [lines 343-353] not lead to other possible outcomes? For example, improve, sharpen or expand the conversation across the divides; increase those committed to the cause?

I am thinking of the likes of John Lewis, and his struggle for racial justice.
Perhaps there could be disagreements, but strategically the cause moves on?
(See Jon Meacham. His Truth Is Marching On: John Lewis and the Power of Hope. First edition. ed. New York: Random House, 2020.)

Author Response

Thanks for your helpful feedback. In response to your comments, I have included the following two paragraphs:

In speaking of religious amnesias, mythologies, and apolitical affects in racist landscapes, this essay does not seek to locate structures of domination solely in white spaces or predominantly white churches. While the category of whiteness includes white spaces and persons, whiteness extends its reach into non-white spaces and persons as well. The essay highlighted the encounter with the Indian American doctor in an Asian-American church setting precisely to make the observation that apolitical affects inflected by whiteness occur in myriad settings.

The doctor’s strong negative reaction to the mention of structural wrongs is significant in light of this essay’s problematization of the felt need to lift up “positive examples” in a discussion about racism. The doctor’s negative reaction downplayed the existence of racialized discrimination. One might ask, could we not raise “positive examples” and still name racism? While such a both-and approach could lead to outcomes other than amnesia and possibly improve, sharpen, or expand the conversation across various racialized divides, the far-reaching depth at which apolitical affects operate in racialized landscapes tempers readily available optimisms.

Reviewer 2 Report

  1. Relating location to affect is a good approach
  2. Would like to see how the argument holds up to non-white centered churches (if there is such a thing) who have accepted and done what the author suggests but still perpetuate the location divide. I.e. move to apolitical neighborhoods and enclaves to have the "comfort" of non-affective "blind" racialized space.
  3. Perhaps some clearers connections could be made not just to the epistemological spaces of the Indian village and racial geography in the US, but also to the physical locations and how they generate affect. What sort of physical restructuring is possible and/or required?

Author Response

(The authors gave the same response as above.)
